# Enhancing Transfer of Reinforcement Learning Agents with Abstract Contextual Embeddings

**Guy Azran**
Technion - Israel institute of technology
guy.azran@campus.technion.ac.il

**Mohamad H. Danesh**
McGill University
mohamad.danesh@mail.mcgill.ca

**Stefano V. Albrecht**
University of Edinburgh
s.albrecht@ed.ac.uk

**Sarah Keren**
Technion - Israel institute of technology
sarahk@technion.ac.il

## Abstract

Deep reinforcement learning (DRL) algorithms have seen great success in performing a plethora of tasks, but often have trouble adapting to changes in the environment. We address this issue by using *reward machines* (RM), a graph-based abstraction of the underlying task to represent the current setting or *context*. Using a graph neural network (GNN), we embed the RMs into deep latent vector representations and provide it to the agent to enhance its ability to adapt to new contexts. To the best of our knowledge, this is the first work to embed contextual abstractions and let the agent decide how to use them. Our preliminary empirical evaluation demonstrates improved sample efficiency of our approach upon context transfer on a set of grid navigation tasks.

## 1 Introduction

Reinforcement learning (RL) algorithms have shown impressive capabilities in a wide variety of problems, from beating video games to robot navigation [1, 2]. The ability to adapt to environment changes is integral in real-world use-cases since they reside in a dynamic world with ever-changing objectives and constraints. However, studies show that popular RL algorithms, in many cases, have difficulty adapting to even the slightest variations in the environment, mainly since they over-fit to the environments on which they were trained [3, 4, 5, 6]. As a result, many new experiences from the altered environment must be sampled to learn new policies even for fairly similar environments.

Consider the following simplified example that depicts two variations, or *contexts*, of a deterministic environment with sparse rewards. A taxi's goal is to pick up a passenger and drop her off at her destination. Only after completing the task does the agent receive a positive reward. Figure 1 depicts two different contexts in this environment. The context in the top figure is seemingly more difficult to solve than that of the bottom figure. Regardless, many RL algorithms trained to optimally solve the environment of the difficult context will still require gathering much data in order to solve the environment of the more simple context, despite the similarities between them.

We aim to improve sample efficiency in *policy transfer*, i.e., to reduce the number of environment interactions required to achieve acceptable performance in a new environment after having interacted with other environments. For this, we introduce *Reward Machine Graph Embeddings as COntext* (RM-GECO) which uses *graph neural networks* (GNNs) [7] to embed a specific type of abstraction, called *reward machines* (RM) [8], into a vector representation. RMs are graphs that represent the structure of the reward function and dynamics of the environment modeled as a Markov decision process (MDP) [9]. RM-GECO utilizes the reward machine GNN embeddings to augment the envi-

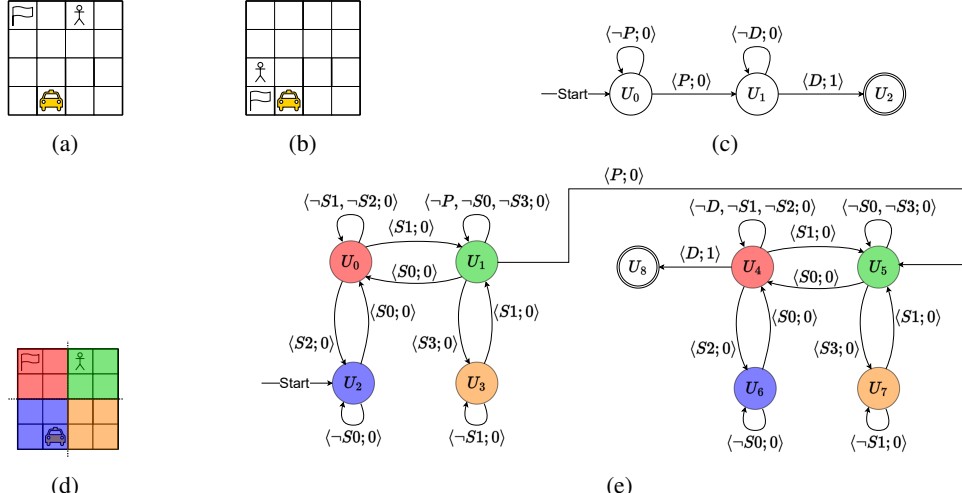

Figure 1: [a,b] Two contexts in the taxi domain. The taxi icon is the acting agent, the person icon is the passenger to be picked up, and the flag icon is the destination. The agent can navigate to any adjacent cell, but cannot cross thick walls. It is aware of its own location and the passenger and destination locations, but not the wall locations. [c] A low resolution reward machine for both contexts presented in with two propositional symbols. P indicates that the passenger has been picked up by the agent and D indicates that the passenger has been dropped off at the destination. Upon initialization, the agent starts in abstract state $u_0$. The agent will not transition to state $u_1$ until P holds true. Similarly, the agent remains in $u_1$ until D holds true, at which time it transitions to terminal abstract state $u_2$. Only when D holds true when in state $u_1$ does the agent receive a positive reward, while any other transition yields a reward of 0. [d, e] For a more detailed RM we split the context of [a] into a smaller $2 \times 2$ grid and use the sectors as new propositional symbols (S0 - S3).

ronment's state space, taken as input by the agent's underlying RL algorithm. By doing so, we give the agent contextual awareness since the context is encoded in the RM.

Prior works improved *transfer learning* in RL, the improvement of learning in a new task through the transfer of knowledge from a related task that has already been learned [10, 11], using various approaches [12, 5]. On the one hand, model-agnostic approaches directly optimize the agent's policy without considering the underlying model. For example, [13] learns individual policies for multiple contexts and aggregates their parameters such that adapting to a new task requires few gradient steps. In contrast, model-based approaches [14, 15] attempt to learn an approximate model of the world that is general enough to capture multiple contexts and can be used to plan behaviors using classical planning algorithms. Our work combines these two approaches by generating an abstract model of the environment, specific to the current context, and providing them as an extra mode of input to model-free RL algorithms. As an abstraction, we use RMs as suggested in [8]. As opposed to [16] that uses the RM states to enhance learning, we embed the entire RM to provide an overall understanding of the task and use it to support transfer learning. We test RM-GECO in multiple environments with different types of variations between contexts. We present preliminary results that support RM-GECO's ability to leverage the model graph topology.

## 2 Background

We support RL agents with partial knowledge of the environment that learn how to operate by acting within it and receiving rewards [17]. Our focus is on *transfer learning* (TL) settings in which agents need to learn a new task through the transfer of knowledge from a previously learned tasks [10, 11].

As is standard in many RL frameworks, the environment is modeled as a Markov Decision Process (MDP) [17] defined as a 5-tuple $M = \langle S, A, T, R, \gamma \rangle$ where $S$ is the state space, $A$ is the action space, $R$ is the reward function, $T$ is the transition distribution function, and $\gamma$ is the reward discount factor. The agent's behavior is determined by a policy $\pi$ such that $\pi(a|s)$ is the

probability of performing action $a$ in state $s$. The *expected return* of policy $\pi$ in $M$ is $J(\pi) = \mathbb{E}_{s_t, s_{t+1} \sim T, a_t \sim \pi(\cdot | s_t)} \left[ \sum_{t=0}^{\infty} \gamma^t R(s_t, a_t, s_{t+1}) \right]$. An optimal policy $\pi^*$ maximizes the expected return.

To model a collection of MDPs across a bounded set of tasks, we use a Contextual MDP (CMDP) [18] which is a collection of *contexts*, each representing an MDP with shared state and action spaces. Formally, a CMDP is a 4-tuple $\langle C, S, A, \mathcal{M} \rangle$ where $C$ is the context space, $S$ and $A$ are the state and action spaces (respectively) shared across all contexts, and $\mathcal{M}$ maps each context to its corresponding MDP, s.t., $\mathcal{M}_c = \langle S, A, T^c, R^c, \gamma \rangle$. Since every context is mapped to a single MDP, we use the terms "context", "MDP", and "environment" interchangeably.

As an MDP abstraction, we use a *reward machine* (RM) [8]. Given a set of propositional symbols $\mathcal{P}$, an RM is a 3-tuple $\mathfrak{R} = \langle U, \delta_u, \delta_r \rangle$ where $U \subset \mathcal{P}$ is the set of abstract states, and $\delta_u : U \times 2^{\mathcal{P}} \to U$ and $\delta_r : U \times U \to \mathbb{R}$ are the abstract transition and reward function respectively, such that $\delta_u(u, p)$ is the next abstract state when in abstract state $u$ and all propositional symbols in $p$ hold true. $\delta_r(u, v)$ is the reward received when transitioning from abstract state $u$ to abstract state $v$.

RM is a form of a finite state machine, and as such can be naturally represented as a graph (see Figure 1c). That is, given an RM $\mathfrak{R}$, we can define it as a graph object $G_{\mathfrak{R}} = \langle V_{\mathfrak{R}}, E_{\mathfrak{R}}, \varphi_{\mathfrak{R}} \rangle$ where:

$$ V_{\mathfrak{R}} = U \qquad E_{\mathfrak{R}} = \{\langle u, v \rangle | \exists p \subset \mathcal{P} \; \delta_u(U, p) = v\} \qquad \varphi_{\mathfrak{R}}(\langle u, v \rangle) = \delta_r(u, v) $$

To represent different contexts with different RMs, we must consider the RM resolution, i.e., the number of RM states. Figures 1d and 1e show how the 3-state RM from Figure 1c is transformed to a finer 9-state RM by considering occupation in a specific section of the map using additional symbols.

We exploit the graphical structure of the RM by embedding it into a deep latent space using GNNs, which are a collection of deep learning algorithms designed to capture information from and perform statistical inference on graph data [19, 7, 20]. This model embeds a graph node by calculating appropriate messages to be sent from its neighbors. The messages themselves are deep representations of two neighboring nodes and their shared attributes, and attempt to capture meaningful information about their relationship. Together, the aggregation of all the messages is all the information the model was able to capture about a specific node in the graph as a function of its immediate neighbors. Recent works using GNNs have shown their ability to capture meaningful information in various learning settings [7, 21, 22, 23]. We expect the messages to capture information useful to determining the current context and how to act within its induced MDP. By composing multiple message passing modules, we can expand the receptive field of the node embedding, providing information from farther nodes, increasing the agent's abstract field-of-view.

## 3 Problem Formulation

We are investigating how RM-based abstractions enhance the ability of an RL agent to transfer to new contexts. The setting we consider is one in which the agent operates in one context or a set of contexts within a CMDP (the *source contexts*), before transitioning to a new *evaluation context*, in which its performance is examined. The source and evaluation contexts are not known a-priori nor is it known to the agent when the transition to the evaluation context will occur.

Let $C = \langle C, S, A, \mathcal{M} \rangle$ be a CMDP and let $\Psi$ be a distribution over $C$. The *transfer learning in contextual MDPs* problem objective is to optimize a chosen transfer utility $U$ in expectation over samples of source and target context sets of given sizes. Among the many ways to measure $U$, we use several measures suggested by [11, 12]. These include jumpstart (JS), time-to-threshold (TT), and transfer ratio (TR). JS measures how well a policy performs on a never-seen-before environment, motivating zero-shot transfer. TT measures how long it takes an agent to achieve acceptable (predetermined) performance, which promotes few-shot transfer. TR measures reward dominance of the policy after transfer over the policy without transfer throughout the training process.

## 4 Reward Machine Graph Embeddings as Contexts

We suggest *RM Graph Embeddings as COntexts* (RM-GECO) that leverages the graph-based abstraction of the MDP offered by the RM to enhance transfer. RM-GECO uses RMs as MDP abstractions and exploits them in two ways. First, as in [8], we perform reward shaping on the RM and provide the resulting rewards to the agent during training instead of the potentially sparse rewards of the original

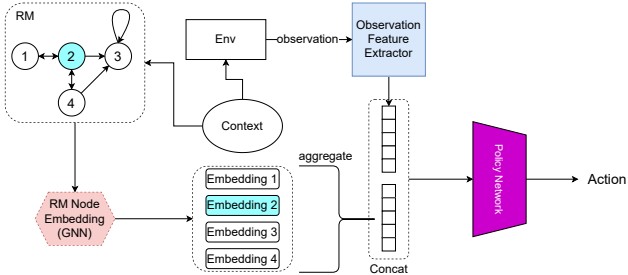

Figure 2: The general architecture of our learning network. The context (center) defines the environment objective and dynamics (top center) and the corresponding RM (top left), The environment observations are processed via an appropriate feature extractor (top right), e.g., CNN, MLP. The RM nodes are embedded into a latent space using a GNN (bottom left). The embedding of the RM node corresponding to the current abstract state (blue) and the observation are concatenated and fed into a policy network (far right) to determine the next action.

MDP. Second, we exploit the RM's graphical structure by augmenting the state space with the RM graph embedding (defined below) and training the agent using the augmented space.

**Definition 1** *A* graph embedding *(GE) function g of rank d is a function that maps graphs to d-dimensional a real-valued vector space, i.e., $g(G) \in \mathbb{R}^d$ for any graph G. Denote $g(G) = \hat{G}$.*

To train the agent, we use deep RL (DRL) methods in which a policy $\pi_\theta$ is implemented as a neural network (NN) with parameters $\theta$. Figure 2 depicts the general framework of our method. We use a GNN $g_\xi$ with parameters $\xi$ as the graph embedding function. The GNN is followed by an aggregation of the resulting node embeddings. To preserve the current abstract state information, we aggregate the node embeddings as a function of the current abstract state, e.g., take only the current abstract state node embedding and ignore the rest.

At every time-step $t$ in context $c$, we embed $G_{\mathfrak{R}_c}$ to $g_\xi(G_{\mathfrak{R}_c}) = \hat{G}_{\mathfrak{R}_c}$ and create augmented state $\hat{s}_t = \langle s_t, \hat{G}_{\mathfrak{R}_c} \rangle$. The policy network $\pi_\theta$ learns the augmented state probability distribution $\pi_\theta(\cdot|\hat{s})$ (instead of $\pi_\theta(\cdot|s)$) (if the environment observations are not represented by vectors, e.g. images, we use an appropriate feature extractor to transform them into vectors). Our training algorithm can be found in Appendix A.

We note that in this work, we focus on examining how RMs can enhance transfer learning, and not on their generation. We therefore assume that for any context-induced MDP $\mathcal{M}_c$, we have a mapping that produces the corresponding RM $\mathfrak{R}_c = \langle U^c, \delta_u^c, \delta_r^c \rangle$. This includes a state-labeling function $L : S \to U$ that maps states to abstract states in the RM. Such mappings can be implemented in various ways, e.g., using factored representations of the state space that extract domain-specific features from sensors [16] or learned from demonstration [24].

## 5 Preliminary Empirical Evaluation

The objective of our empirical evaluation is to demonstrate the benefit of RM-GECO in enhancing our evaluated transfer utilities. Our preliminary evaluation was performed on a 4x4 grid-navigation setting in which an agent randomly spawns within the map and must navigate through adjacent cells to a specified location that depends on the current context. The agent receives a reward of +100 upon completion of its objective and a reward of -1 at every step taken. Additional information about the environment and evaluation is available in Appendix B and C.

Our results show promise in RM-GECO's ability to represent context and enhance an agent's transfer learning capabilities. The most significant improvement occurs when the agent is not aware of the current context until the RM embeddings are introduced. We further see slight improvement in settings where agents are context aware, i.e., the context is a part of the observation space.

Figure 3 shows evaluation results throughout training on the source contexts and after transfer to the target context using three RM resolutions. The results reveal several phenomena. Firstly, using GE

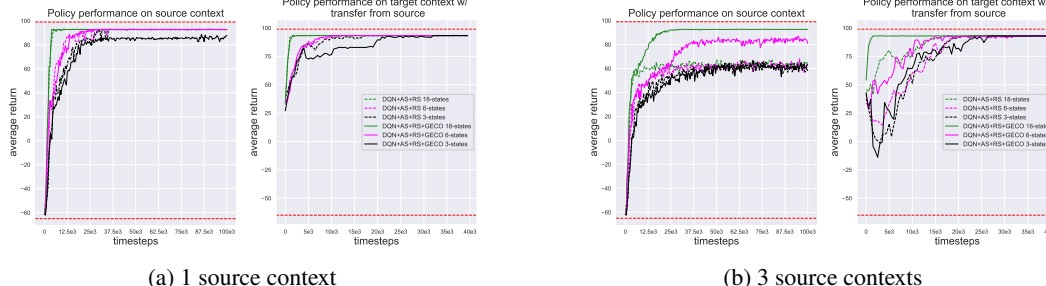

(a) 1 source context  (b) 3 source contexts

Figure 3: Average return over number of timesteps (interactions with the environemnt) throughout training in the GN environment with a 4x4 grid under the changing map obstacles context space, averaged over 15 separate trials. Each subfigure contains results for training on all source contexts (left) and on a single unseen target context. Subfigures differ by the number of source contexts to which the agent is exposed before transfer (results on more source context are provided in the Appendix D). Different colors represent different RM resolutions. The RM states take the form of a 1x1 (low-res), 2x2 (mid-res), or 4x4 (high-res) grid split of the taxi domain to create RMs of 3 (green), 6 (blue), and 18 (red) abstract states respectively. Dotted lines represent DQN with AS and RS modifications as in [16], and solid lines have an additional GECO modification (ours)

when training on a single source context is on-par or slightly worse that not using them at all. This occurs because the reward machine does not change throughout training on the source task, making this constant input worth ignoring. Likewise, using GE of the low resolution RM has only a mild affect on performance since this RM is shared between all contexts, and thus does not capture any contextual changes. Finally, when training on multiple source contexts using GE with RMs capable of capturing contextual changes, we are able to achieve higher returns, which has a direct affect on transfer performance. The high resolution RMs are unique to each context since they represent each cell of the grid with its own abstract state (i.e., $|U| = |S|$). Thus, it is always able to find an optimal policy, and improves transfer performance significantly as the number of source contexts increases, providing near zero-shot performance with 10 contexts. Each mid resolution RM represents a subset of contexts. As the number of source contexts grows, the number of contexts that overlap the same RM grows, creating confusing experiences (as in the low-res RM, but less severe). Results for further experimentation can be found in Appendix D.

# 6   Conclusion and Next Steps

We present a novel approach to enhancing transfer learning of RL agent using deep latent embeddings of graph-based abstractions, called reward machines. Our experiments demonstrated the benefit of our approach on transfer on a limited set of grid-navigation benchmarks. While initial results are promising, we believe our approach will yield greater benefit in more complex tasks.

Throughout this paper we mention environment "complexity" without formally defining it. Since we expect our model to adapt to complex tasks, we intend to suggest formal definitions of task complexity and explore the benefit of our approach on increasingly complex settings. Using this definition, we intend to extend our evaluation to more complex domains.

As another future objective, we wish to better understand the policy's deep latent space and how it is affected by using our method. This includes the GNN output and the policy penultimate layer. We wish to project these latent vectors to a low-dimensional space to find patterns, clusters, and anomalies that will provide insights into the actual benefits of our representations. As a part of this analysis, we will explore and compare additional node embedding and aggregation methods.

Finally, as our results show, using graph embeddings can delay convergence to the optimal policy when training in the source contexts due to the larger number of learnable parameters. To overcome this, and improve transfer performance, we plan to examine the effect of pre-training the GNN will have on convergence.

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

# A  Algorithm

---

**Algorithm 1** Training RM-GECO with DQN

---

    **Input**: $\pi = \langle g_\xi, Q_\theta \rangle$ - RM-GECO policy for DQN;
    **Input**: $C$ - contexts set;
    **Input**: $N$- number of episodes to train;

1:  $\mathcal{D} \leftarrow$ empty experience replay buffer
2: **for** $i \in [N]$ **do**
3:     $c \leftarrow sampleUniform(C)$
4:     **for** each step in episode of $\mathcal{M}_c$ **do**
5:         $\hat{s} \leftarrow \langle s, g_\xi(G_\Re) \rangle$                     ▷ Work with augmented state space
6:         $q \leftarrow Q_\theta(\hat{s})$
7:         $a \leftarrow \epsilon\text{-greedy}(q)$
8:         $s' \leftarrow \mathcal{M}_c.\text{step}(a)$
9:         $v \leftarrow \delta_u^c(u, L(s'))$
10:        $r \leftarrow \delta_r^c(u, v)$                           ▷ Using RM reward
11:        $\hat{s}' \leftarrow \langle s', g_\xi(G_\Re) \rangle$
12:        $\mathcal{D}.\text{store}(\hat{s}, a, r, \hat{s}')$
13:        $\hat{s}, a, r, \hat{s}' \leftarrow \mathcal{D}.\text{sampleBatch}()$
14:        $l \leftarrow \sum\limits_{\text{batch}} (r + \max_{a'} \{Q_\theta(\hat{s}')[a']\} - Q_\theta(\hat{s})[a])^2$
15:        $\xi, \theta \leftarrow \nabla_{\xi,\theta} l$
16:        $s, u \leftarrow s', v$
17:     **end for**
18: **end for**
19: **return** $g_\xi, Q_\theta$

---

Algorithm 1 is an example implementation of training our method using deep Q-learning [1] (DQN) as the underlying policy algorithm with initial policy $\pi$ (comprised of a parameterized graph embedding $g_\xi$, and action-value function $Q_\theta$). It differs from DQN in two ways. First, it samples a new context at the end of every episode, i.e., until the task is completed, the agent has failed, or the agent has reached a cap number of steps in the environment. Second, we augment the state space with a RM graph embedding. Lastly, we ignore the environment's reward and use the RM reward as in [8, 16]. We note that the choice of DQN is arbitrary, and any vector-based DRL algorithm can be adapted to use RM-GECO.

# B Evaluation Baselines

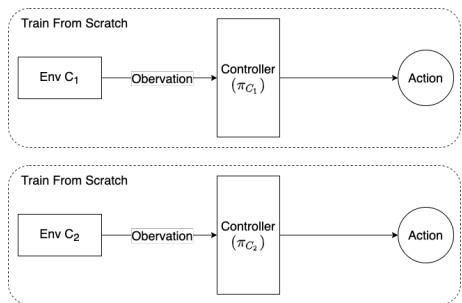

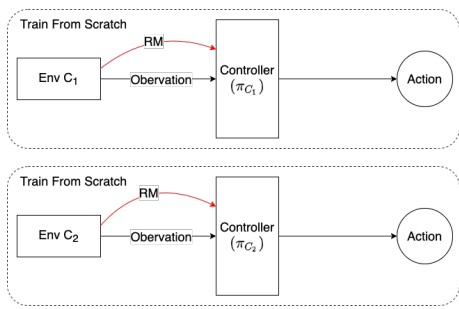

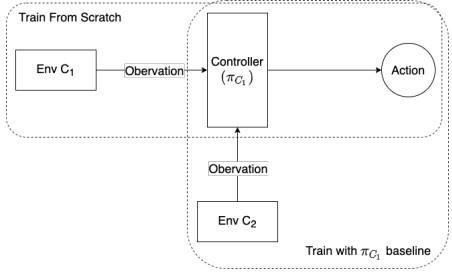

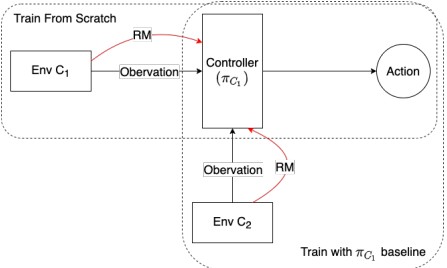

(a) Training a policy from scratch on $C_2$ without using the policy from $C_1$

(b) Training a policy with reward machine input and reward shaping from scratch on $C_2$ without using the policy from $C_1$.

(c) Policy transfer from $C_1$ to $C_2$

(d) Policy transfer from $C_1$ to $C_2$ with individual reward machines per context

Figure 4: Our evaluation baselines setup. $C_1$ is the training context on which we train the policy until convergence. $C_2$ is a second context to which we are trying to adapt with minimal sample complexity
.

## B.1 Setup: No Transfer

**Training session** We start with a randomly initialized policy network $\pi$. Given a context $c$, we train $\pi$ using the chosen DRL algorithm, e.g. DQN, on environment $\mathcal{M}_c$. Training ends once we have performed a predetermined number of steps within the environment. After training, we output the trained policy, denoted $\pi_c$.

**No transfer** We test here our method's ability to sustain a low TT when training on the source context, i.e., training from scratch. For every context $c$ we train our policy $\pi$ on $\mathcal{M}_c$, yielding a trained policy $\pi_c$, while evaluating the policy during training every few iterations as a deterministic policy. After training, we evaluate TT.

## B.2 Setup: Transfer

**Transfer learning session.** In this setting we are given a source contextSet, $C_{\text{src}}$ and a target context set $C_{\text{tgt}}$. As in the previous subsection, we start with a randomly initialized policy network $\pi$ and train it with the chosen DRL algorithm on $\mathcal{M}_{C_{\text{src}}}$ for a predetermined number of steps time-steps to produce $\pi_{C_{\text{src}}}$. We then train $\pi_{C_{\text{src}}}$ in environment $\mathcal{M}_{C_{\text{tgt}}}$, yielding $\pi_{C_{\text{src}},C_{\text{tgt}}}$, the transfered policy.

**Transfer.** During training, we evaluate the policy (as a deterministic policy) at constant intervals. When training is complete, we calculate $U_{JS}$ (zero-shot transfer), $U_{TT}$ (few-shot transfer), and $U_{TR}$ (transfer dominance). We repeat this experiment with changing the source context set sizes on which the policy is trained before transfer.

**Configurations** We train DQN [1] and DQN [25] with combinations of several modifications. The modifications we consider are the following:

- Abstract State (AS) - the current abstract state in the RM is provided as input.
- Reward Shaping (RS) - replaces the environment reward with the reshaped RM rewards.
- Graph Embeddings (GECO) - embeds the RM graph and augments the state space.

### B.3 Setup: RM Resolution

To understand the effect of abstraction resolution on our performance measures, we repeat our experiments with three different RM resolutions. One is to be as vague as possible such that most contexts will have the same RM. The second is a very high resolution abstraction that captures all or most of the actual MDP we are solving. The third resolution is somewhere in the middle between the first two. Figure 1 shows an example low-resolution and mid-resolution RMs. A high resolution RM would capture each cell of the grid as an abstarct state.

## C   Evaluation Environments

We test our method in environments with complex tasks and sparse rewards. For this, we developed194 the Multi-Taxi environment as an extension of open-AI's single taxi domain Brockman et al. [2016].195 (Figure 1 depicts the environment in the single-agent setting) A group of taxis is tasked with picking up and dropping off all passengers at their destination. They have individual observation types that can either be a symbolic vector describing the state, an image of the environment rendering, or both. Possible actions include navigating in one of four directions, pickup and dropoff of a passenger, refueling, and standby (no action). Positive reward are received upon completing the task.

The environment's configurable features allow the user to set the number of passengers and taxis, the taxi's capacity and fuel requirements, the actions' stochasticity, the sensor function, and more. We note that while multi-taxi is natively a multi-agent environment, we explore it as a single-agent setting. By leveraging the domain's customizability we define four environment settings of different complexity levels. For each setting, we specify the differences between the contexts in the CMDP.

**Grid Navigation (GN):**   The agent must navigate to a single passenger. This is equivalent to navigating to a particular cell in a grid and performing a "done" action when the destination cell has been reached.

**Apple Picking (AP):**   The agent receives a positive reward when picking up all passengers (apples), among the multiple passengers that are distributed in the environment. There is no need to drop-off the passengers. Here, we can regulate the complexity of the task, but there are no long-term dependencies between actions, unlike pickup and drop-off.

**Pickup and Drop-off (PD):**   The taxi receives a positive reward when delivering passengers from their starting locations to their destinations. We consider both the single passenger and multiple passenger settings. The task imposes long-term dependencies in that a picked up passenger must reach its destination before reward is collected.

**PD with Fuel Constraints (PDFC):**   An extension of the pickup and drop-off setting where the taxi has a limited amount of fuel. The taxi may need to refuel at a station to achieve the goal. This adds another layer of complexity to the previous settings and may cause severe future consequences to performing irrelevant actions.

## D   Additional Results

The objective of our empirical evaluation is to demonstrate the benefit of RM-GECO on transfer. Our preliminary evaluation is on a 4x4 grid-navigation setting. Our first experiment is more of a sanity check to see if the GNN embedding is able to capture information relevant for policy learning. In it, we compare modifications AS +RS [16] to our suggested solution modifications AS +RS +GECO (RM-GECO) with DQN in the context of changing map obstacles. The map context is special because the change is not embodied in the observation space, making it impossible to find an optimal

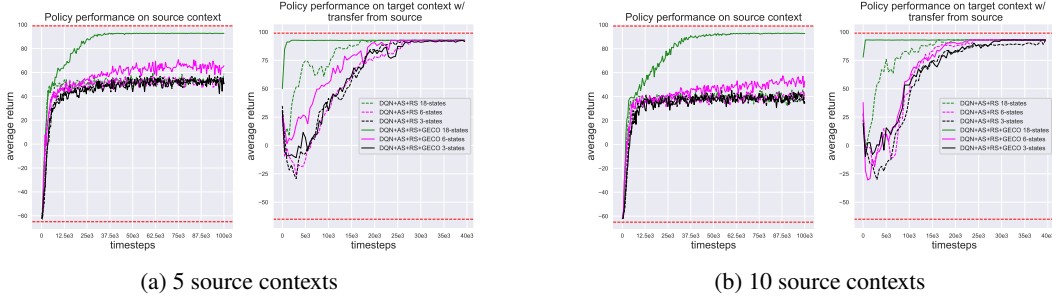

(a) 5 source contexts                                          (b) 10 source contexts

Figure 5: Results on more contexts in settings similar to 3.

policy for multiple contexts. Thus, using our RM reveals information that is otherwise unattainable and uninferrable. Our results show that the GNN is in fact learning useful representations for this navigation problem that allow for better performance on multiple contexts even though the original observation is incomplete.

Figure 5 provides results on more experiments throughout training on the source contexts and after transfer to the target context using three RM resolutions. Similar conclusions to Section 5 can be made.

Figure 6 shows agent performance throughout the training process. They compare the average return over increasing time-steps of standard DQN [1] and DQN using RM-GECO with a 3-state (Figure 6a), a 6-state (Figure 6b), and an 18-state (Figure 6c) RM. Each sub-figure contains a source context plot (left) and a target context plot (right). The source context plots contain the average return achieved over the course of training in the source context from scratch. Similarly, the target context plots show results for the target context using the learned policies from the source context.

Looking at the source context plots, we see a delay in convergence to an optimal policy. This is true for all three RM resolutions, but is more significant in the higher resolution RM. That is to be expected due to the larger number of learned parameters in RM-GECO. However, we notice a slight boost in transfer performance for both JS and TT when training on more than one context.

GNN node embeddings are aggregated to a single fixed-length vector with which we augment the state space (see Figure 2). The results we show in Section 5 all use the same aggregation which takes the current abstract state node embedding and ignores the rest of the node embeddings. To further analyze this, we ran an additional experiment in the changing map context space with a different aggregation which takes the mean node embedding and ignores the current abstract state. The results show that both aggregation methods share similar performance in transfer, but that the current state abstraction shows faster convergence on the source task, when dealing with higher resolution RMs.

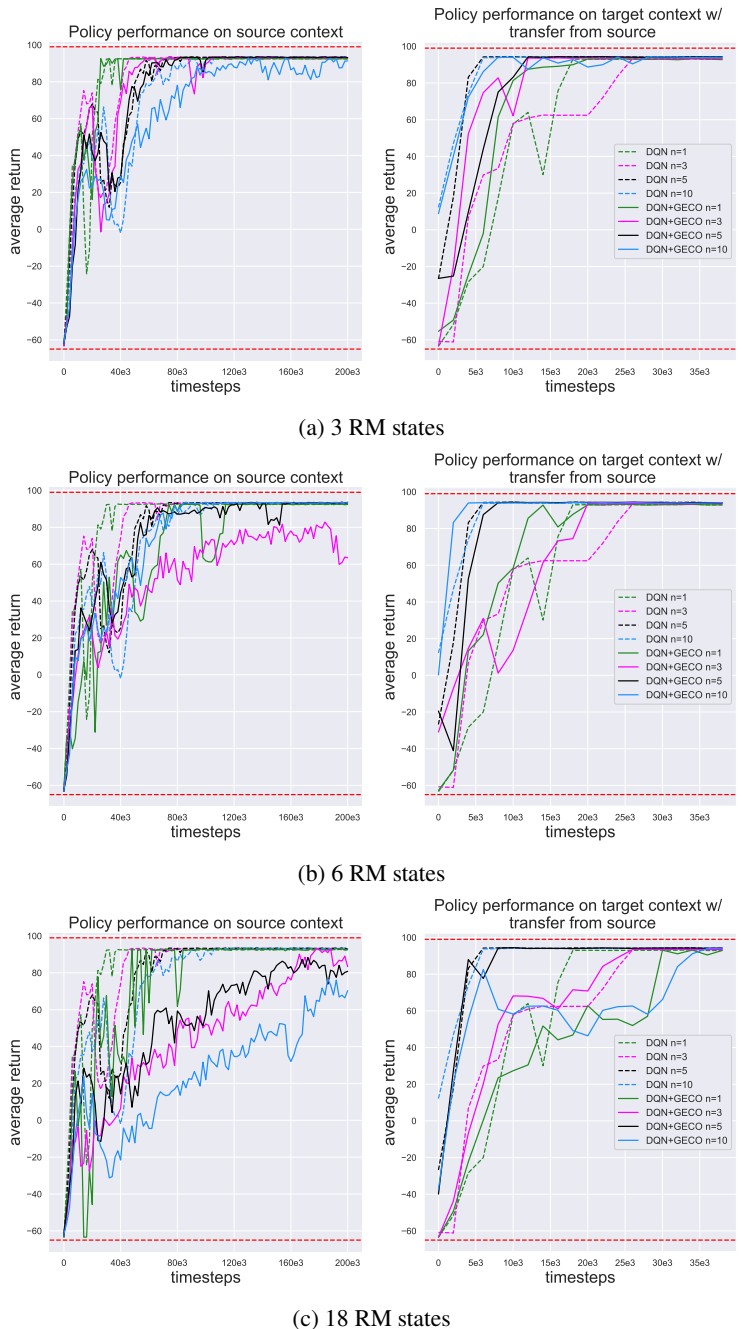

Figure 6: Average return over number of timesteps (interactions with the environment) throughout training in the GN environment with a 4x4 grid under the changing entities context space, averaged over 5 independent runs. Each color in the plot represents the number of source contexts to which the agent is exposed before transfer, denoted n. Dotted lines represent standard DQN while solid lines use GECO. The left plot shows evaluation on the source contexts and the right plot shows evaluation on the target context after transfer from the source context.

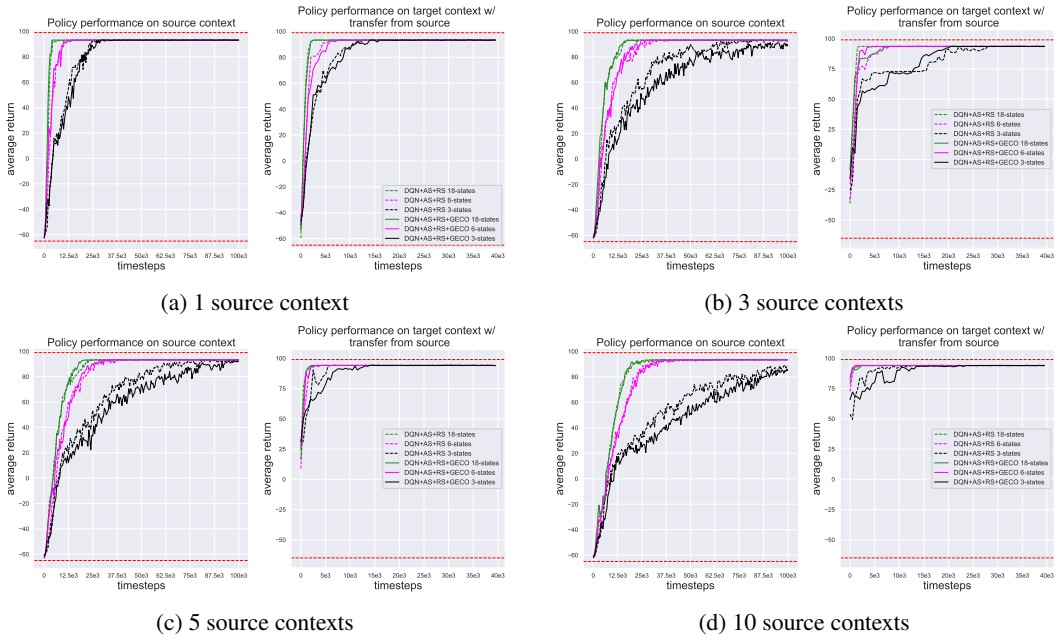

(a) 1 source context

(b) 3 source contexts

(c) 5 source contexts

(d) 10 source contexts

Figure 7: As in Figure 3 for the fixed entities context space.

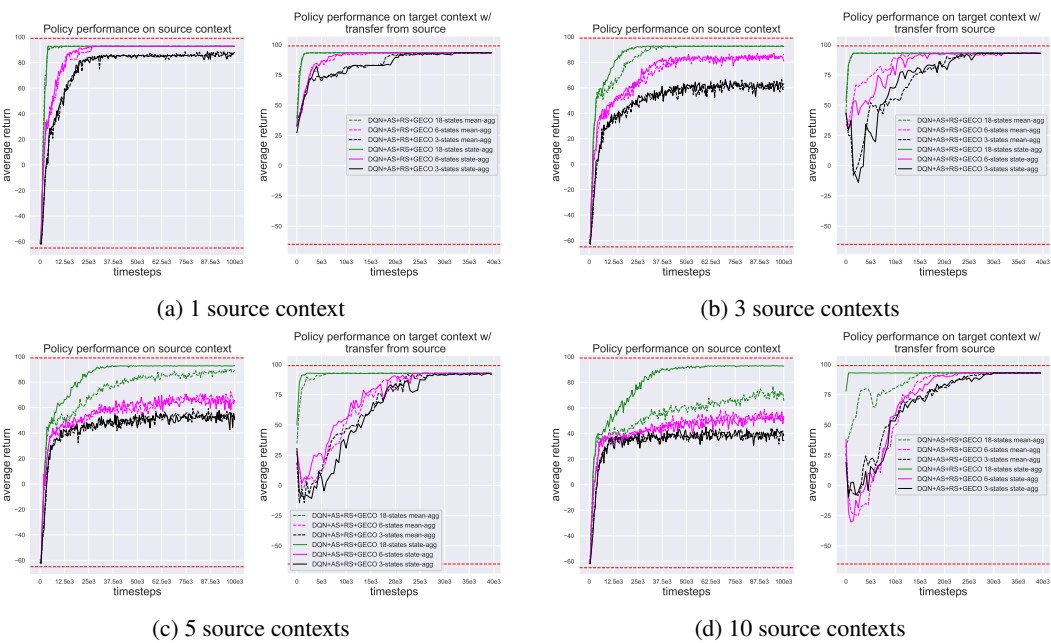

(a) 1 source context

(b) 3 source contexts

(c) 5 source contexts

(d) 10 source contexts

Figure 8: As in Figure 3, but comparing the GECO using the "current state" and "mean" aggrerations. Solid lines correspond to the current state aggregation and dashed lines to the mean aggregation.

