# OpenReview forum: "Enhancing Transfer of Reinforcement Learning Agents with Abstract Contextual Embeddings"
_NeurIPS.cc/2022/Workshop/nCSI — nCSI WS @ NeurIPS 2022 Poster_

### Official Review · Reviewer_8wLk · 2022-10-11
**Review of Enhancing Transfer of Reinforcement Learning Agents with Abstract Contextual Embeddings**

**Rating:** 2
**Confidence:** 3

**Review:**

**Summary**

This paper proposes to use the abstract reward machine (RM) as an additional context to learn transferable RL policies. The proposed approach learns a graph embedding of the RM, called RM-GECO, and concatenates the node embedding of the current state to the environment observation before learning the policy. The paper hypothesizes that the RM-GECO provides additional information about the context that helps in generalizing the RL policies to different tasks. The preliminary experiments aim to evaluate this hypothesis on grid navigation tasks.


**Strengths**
* The proposed idea of using RM as context is quite novel.
* The proposed approach is relevant for the nCSI workshop as RM may encode dynamic causal information of the domain.


**Weaknesses**
* The paper proposes the use of RM-GECO to learn policies and compares the proposed approach to learning without any context. While the selected baseline is important, it is not sufficient. To justify the use of RM-GECO, a baseline policy that uses some other information as context is essential. Perhaps something like policy sketches [1], or the use of RM without embedding.
* It is not clear how the preliminary experiments demonstrate the benefit of their approach. How is DQN+GECO better than DQN in Figure 3?
* The paper needs more evaluation and analysis of when would the RM-GECO help?

**Questions**
* Figure 3 shows evaluations on a 4x4 grid using RM with 6 and 18 states. How do these curves look when the number of RM states is 3 (like in Figure 1c)? A more thorough discussion of the required resolution of the RM would help.

---

### Official Review · Reviewer_wYWi · 2022-10-15
**Interesting concept and good formulation. The results are a bit weak without additional modifications.**

**Rating:** 2
**Confidence:** 2

**Review:**

The idea to use contextual embeddings for a more efficient task-transfer is interesting. The results of DQN+GECO do not show a significant improvement over the baseline. However, the experiments from appendix where they do DQN+AS+RS+GECO is much more convincing. I'm unsure why the plots from the appendix with the additional modifications weren't shown in the main paper.

## Pros
- The idea of adding contextual graph embeddings for task transfer is quite interesting.
- It brings a better understanding of the world to the agent.
- The mathematical formulation and problem setup is discussed well.
- Using GECO along with modifications like RS (rewards shaping), AS (abstract spaces) and state-aggregation seems to be making a big improvement.

## Cons
- Using just GECO alone don't show the convincing effectiveness that the method brings. This might be due to the choice of the task which is relatively less-complex, or due to the sparse rewards from the environment (without reward shaping).
- This might change in more complex tasks, as the authors mentioned. However, then, my concern would be the increased RM states that comes with it. It would be interesting to see how they'd handle the resolution during that case.
- The work would be more sound once the authors show the performance results from the future tasks that they mention, e.g. using a pre-trained GNN or using an auxiliary loss.


A side note: Please increase the figure size to make them more readable.

---

### Meta-Review · Area_Chair_UrYo · 2022-10-18

**Recommendation:** 2
**Confidence:** 4

**Metareview:**

The reviewers have done written reasonably insightful reviews. I request the authors to improve the exposition to clarify the specific questions by the reviewers -- specifically on experiments including but not limited to the baselines.

---

### Decision · Program_Chairs · 2022-10-20

Accept (Poster)